# Performance Study of a New Cluster Splitting Algorithm for the Reconstruction of PANDA EMC Data

**Ziyu Zhang [1], Guang Zhao [2,*], Shengsen Sun [2,3], Qing Pu [1], Chunxiu Liu [2], Chunxu Yu [1], Dong Liu [4], Hang Qi [4], Guangshun Huang [4], Tobias Stockmanns [5], Beijiang Liu [2], Fei Wang [6], Yitong Zhang [7] and Xiaoyan Shen [2,3]**

1 School of Physics, Nankai University, Tianjin 300071, China
2 Institute of High Energy Physics, Beijing 100049, China
3 School of Physical Sciences, University of Chinese Academy of Sciences, Beijing 100049, China
4 Department of Modern Physics, University of Science and Technology of China, Hefei 230026, China
5 Forschungszentrum Jülich, Institut für Kernphysik, 52428 Jülich, Germany
6 School of Nuclear Science and Technology, University of South China, Hengyang 421001, China
7 School of Physics, Liaoning University, Shenyang 110036, China
* Correspondence: zhaog@ihep.ac.cn

**Abstract:** For high-energy $\pi^0$ mesons, the angle between the two final-state photons decreases with the increase in the energy of the $\pi^0$, which enhances the probability of overlapping electromagnetic showers. The performance of the cluster splitting algorithm in the EMC reconstruction is crucial for the mass resolution measurement of $\pi^0$ with high energy. The cluster splitting algorithm is based on the theoretical lateral distribution of the electromagnetic showers. A simple implementation of the lateral distribution can be described as a (multi-)exponential function. In a realistic electromagnetic calorimeter, considering the granularity of the detector, the measured energy in a cell is actually the integral of the theoretical energy deposition, which deviates from the exponential function. Based on the simulation of the barrel EMC in the $\overline{P}$ANDA experiment, a cluster splitting algorithm with a new lateral energy development function is developed. The energy resolution of overlapping showers with high energy has been improved.

**Keywords:** calorimeter; energy reconstruction; cluster splitting algorithm

## 1. Introduction

The antiProton ANnihilations at DArmstadt ($\overline{P}$ANDA) experiment [1,2] is planned to operate in 2026 at the Facility for Antiproton and Ion Research (FAIR). $\overline{P}$ANDA aims to perform the studies of charmonium spectroscopy, exotic states, charmed hadrons in nuclear matter, and the $\gamma$-ray spectroscopy of excited states in doubly strange $\Lambda\Lambda$ hypernuclei with the beam momentum in the range from 1.5 GeV/c to 15 GeV/c.

In order to ensure a geometrical acceptance close to $4\pi$, the $\overline{P}$ANDA detector consists of two spectrometers: the target spectrometer (TS) and the forward spectrometer (FS), as shown in Figure 1 [1]. The TS is arranged in a barrel part for angles larger than $22°$ and an end cap part for the forward range down to $5°$ in the vertical and $10°$ in the horizontal plane, while the FS covers the very forward angles. Both the TS and FS contain detectors for tracking, charged particle identification, electromagnetic calorimetry, and muon identification.

The electromagnetic calorimeter (EMC) is an essential detector in $\overline{P}$ANDA to measure the energy of the photons and electrons. The EMC in the TS consists of one barrel and two end caps. To achieve the desired very low detection threshold, the improved $PbWO_4$ scintillator was chosen, providing a small radiation length of 0.89 cm and a short decay time of 6.5 ns. The calorimeter will be operated at a temperature of $-25\ °C$ for an increased light yield [3]. There are 11200 crystals in total for the barrel EMC, with an average lateral size of 21.3 mm. The barrel is approximately projective. The crystals almost focus on the collision vertex with a $4°$ tilt in both the azimuthal and longitudinal directions.

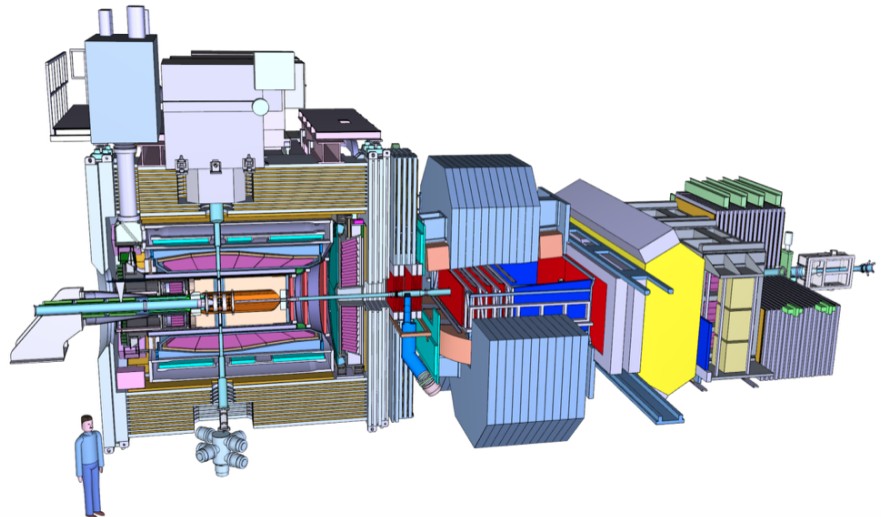

**Figure 1.** Side view of $\overline{P}$ANDA with the target spectrometer (TS) on the left side and the forward spectrometer (FS), starting with the dipole magnet center, on the right side. The antiproton beam enters from the left.

The main task of the EMC is to provide the four-vector momentum of a photon by measuring its energy and position. When photons overlap, the EMC should also be able to isolate the overlapped photons, especially in the case of a high-energy $\pi^0$. Figure 2 shows the angle between the two final-state photons from $\pi^0$ decays as a function of the $\pi^0$ energy. The overlapping is much more severe with the increase in energy. Therefore, any significant improvement in cluster splitting can improve the $\pi^0$ reconstruction at high energy.

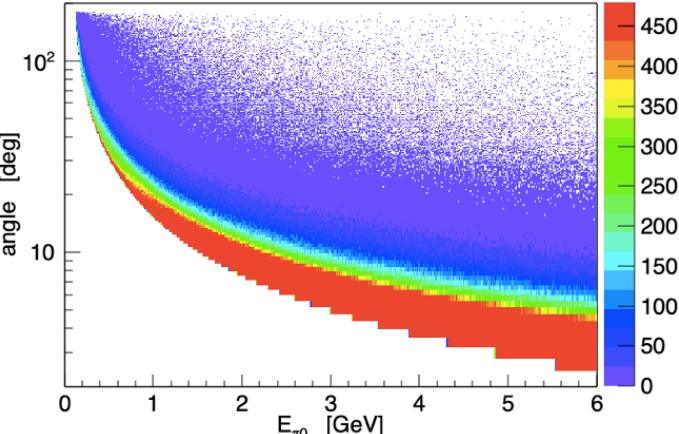

**Figure 2.** The angle between the two final-state photons with the increase in the energy of $\pi^0$.

In this paper, an optimized cluster splitting algorithm is presented, applying an updated description of the lateral development formula of electromagnetic showers that takes the detector granularity into account.

## 2. Cluster Splitting Algorithm

The two major procedures of the EMC reconstruction are cluster finding and cluster splitting. The cluster finding algorithm attains the objective of finding a cluster formed by a series of crystals with an energy deposition higher than the threshold in a continuous region. In a cluster, a seed crystal is defined as the crystal with local maximum energy. If there are multiple seeds in a cluster, it means shower overlaps exist, and it should be further separated. The cluster splitting algorithm is intended to separate and precisely assign the energy and position of each overlapped shower.

The cluster splitting algorithm is based on the knowledge of the lateral development of electromagnetic showers, which can be expressed as:

$$\frac{E_{\text{target}}}{E_{\text{seed}}} = \text{LAT}(r), \tag{1}$$

where $E_{\text{target}}$ and $E_{\text{seed}}$ are energy depositions of the target crystal and seed crystal, $r$ is the distance from shower center to the target crystal, and LAT is a function of $r$ describing the lateral development. As shown in Figure 3, for a target crystal, the energy deposition fraction from the $i$-th shower can be calculated as:

$$f_i = \frac{(E_{\text{seed}})_i \cdot \text{LAT}(r_i)}{\Sigma_j (E_{\text{seed}})_j \cdot \text{LAT}(r_j)}, \tag{2}$$

where the sum in the denominator runs over all showers in the cluster. With the energy deposition fraction $f_i$, the energy deposition from different showers in a crystal can be separated, and the energy and position of the individual shower can be calculated. The energy fraction will be updated iteratively according to the energy and position of the showers in the previous step until the results converge.

- 🟢 the target crystal
- 🔵 the seed crystal
- 🟡 the shower center

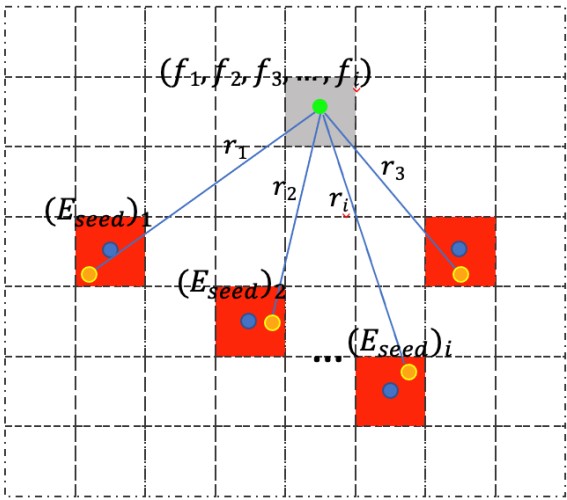

**Figure 3.** Demonstration of calculating the energy deposition fraction from seeds (red) for a target crystal (grey). $f_1, f_2, f_3$, and $f_4$ are energy fractions; $(E_{\text{seed}})_1$, $(E_{\text{seed}})_2$, $(E_{\text{seed}})_3$, and $(E_{\text{seed}})_4$ are different seed crystal energies; and $r_1, r_2, r_3$, and $r_4$ are the distances from the center of showers to center of target crystals.

### 3. Lateral Energy Distribution of Electromagnetic Shower Considering the Detector Granularity

The lateral development distribution of an electromagnetic shower can be empirically described as an exponential function (shown by the black dashed line in Figure 4):

$$\text{LAT}^{(\text{E})}(r) = \exp(-c \cdot r / R_{\text{M}}), \tag{3}$$

where $R_{\text{M}}$ is the Molière radius of PbWO$_4$, and the $c$ is a constant. For a realistic electromagnetic calorimeter geometry, the energy deposition in a crystal is the integral of Equation (3) over the crystal area, which deviates from the exponential function. Based on the $\overline{\text{P}}$ANDA barrel EMC, a measurement of the lateral development is performed using MC-simulated events. The results of the simulation are shown by the scattered points

in Figure 4. An updated parameterization for the lateral development that considers the detector granularity is proposed as:

$$\text{LAT}^{(G)}(r) = \exp\left(-\frac{p_1}{R_M} \cdot \left(r - p_2 \cdot r \cdot \exp\left(-\left(\frac{r}{p_3 \cdot R_M}\right)^{p_4}\right)\right)\right), \tag{4}$$

where $p_1, p_2, p_3$ and $p_4$ are free parameters. Equation (4) can be viewed as adding a correction term for the small $r$ region to the original exponential Equation (3).

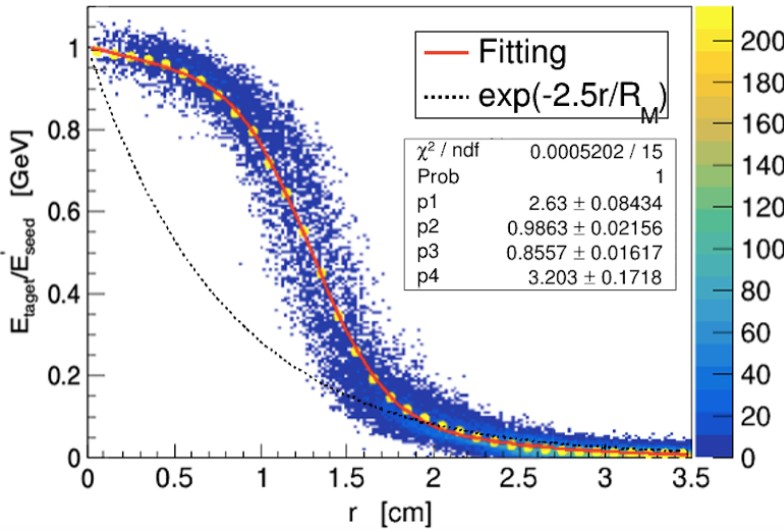

**Figure 4.** The $E_{\text{target}}/E_{\text{seed}}$ varies with the $r$. The $r$ is the distance from the shower center to the target crystal. Each point in the scatter plot represents a target crystal in the showers with the energy $E_{\text{target}}$. The black dashed line is the empirical exponential function; the red line represents the fitting analytic function.

With the updated lateral development model, a correction on the $E_{\text{seed}}$ is further applied. When the shower center does not coincide with the center of the seed crystal, the actual seed is a virtual seed that slightly deviates from the physical crystal. As shown in Figure 5, if we consider a virtual seed as the yellow point, the virtual seed energy ($E'_{\text{seed}}$) can relate to the seed crystal energy ($E_{\text{seed}}$) as:

$$E_{\text{seed}} = E'_{\text{seed}} \cdot \text{LAT}^{(G)}(r_{\text{seed}}), \tag{5}$$

where $r_{\text{seed}}$ is the distance from the center of the shower to the geometric center of the seed crystal. The lateral development distribution formula is updated with the seed energy correction:

$$\text{LAT}^{(G)}_{\text{corr}}(r) = \exp\left(-\frac{p_1}{R_M} \cdot \left(\xi(r, p_2, p_3, p_4) - \xi(r_{\text{seed}}, p_2, p_3, p_4)\right)\right),$$
$$\text{where } \xi(r) = r - p_2 \cdot r \cdot \exp\left(-\left(\frac{r}{p_3 \cdot R_M}\right)^{p_4}\right). \tag{6}$$

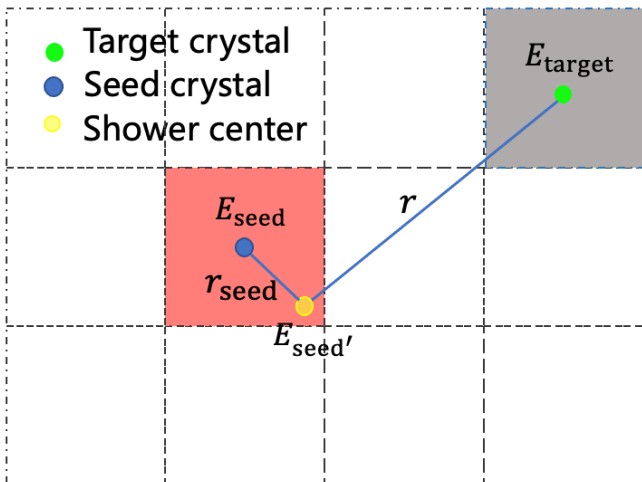

**Figure 5.** This figure shows a case in which the center of the shower is inconsistent with the center of the seed crystal. The green point is the center of the target crystal, and the blue point and yellow point represent the geometric center of the seed crystal and the center of the shower, respectively.

## 4. Performance Results

The performance of the updated cluster splitting algorithm that considers the detector granularity has been checked using small cross-angle photon samples. Samples of two photons with the same energy (6 GeV) and small opening angle at which the photons are emitted (<6.75°) are simulated. The small open angle ensures overlapping between the two electromagnetic showers on the $\overline{\text{P}}$ANDA EMC. Figures 6 and 7 show the reconstructed energy distributions. The black and red histograms represent the energy of photons reconstructed by the default algorithm and the updated algorithm, respectively. The comparison of two histograms demonstrates intuitively that the energy reconstruction of photons is better using the updated algorithm. For a more precise comparison, the histograms have been fitted by double-Gaussian functions. From the fitting results, the energy resolutions are obtained as $199.1 \pm 2.6$ MeV for the default algorithm and $156.2 \pm 1.3$ MeV for the updated algorithm. For 6 GeV overlapping showers, a roughly 20% improvement is achieved on the energy resolution with the updated cluster splitting algorithm.

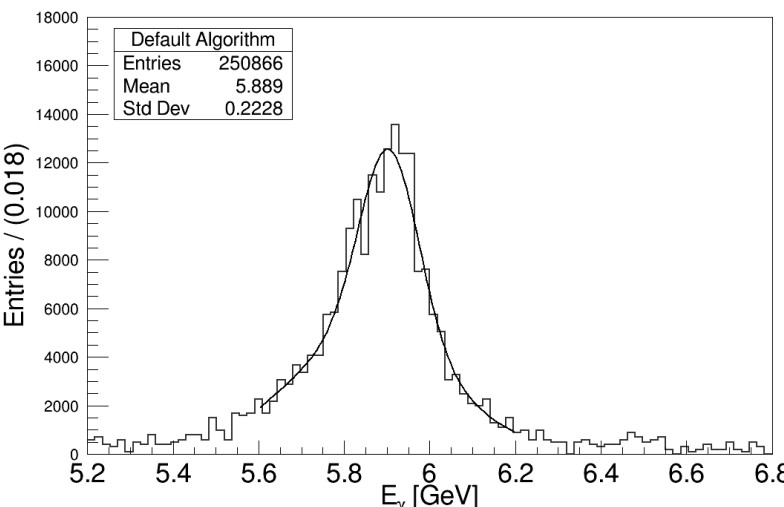

**Figure 6.** Reconstructed photon energy distribution with the default algorithm.

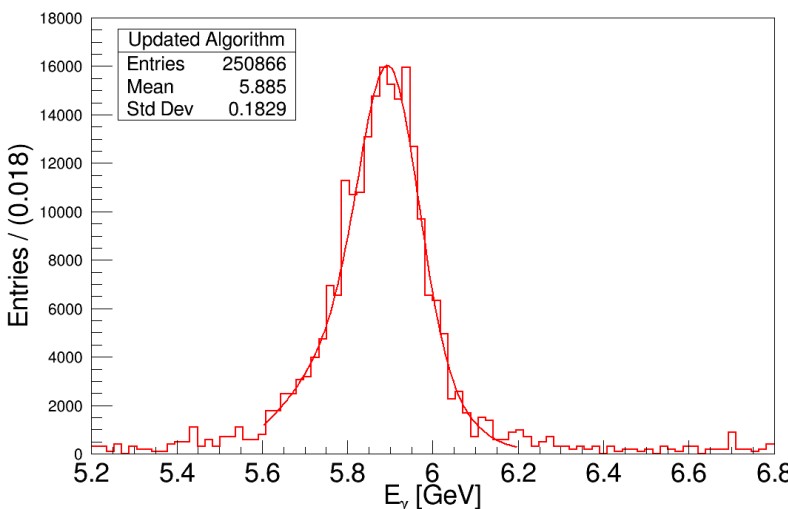

**Figure 7.** Reconstructed photon energy distribution with the updated algorithm.

## 5. Summary

The cluster splitting algorithm has been updated by establishing a new description of the lateral development of an electromagnetic shower for the barrel $\overline{\text{P}}$ANDA EMC. The performance is studied in a small cross-angle photon MC-simulated sample. A clear improvement in the energy resolution is achieved for high-energy photons, which may lead to improvement in the $\pi^0$ mass resolution. More validation of the updated cluster splitting algorithm will be carried out in the future. The algorithm can also easily be applied to other calorimeter sub-detectors in $\overline{\text{P}}$ANDA.

**Author Contributions:** Conceptualization, G.Z. and S.S.; methodology, Z.Z., Q.P., G.Z., S.S., and T.S.; software, Z.Z., Q.P., C.L., D.L., H.Q., and T.S.; validation, Z.Z., F.W., and Y.Z.; formal analysis, Z.Z., Q.P., and G.Z.; investigation, Z.Z., S.S., G.H., and B.L.; writing—original draft preparation, Z.Z. and G.Z.; writing—review and editing, G.Z. and S.S.; supervision, C.Y., G.H., and X.S.; project administration, S.S. and X.S. All authors have read and agreed to the published version of the manuscript.

**Funding:** This research was funded by National Natural Science Foundation of China (NSFC) under contract nos. 11875277 and 12061131003 and the National Key R&D Program of China under contract no. 2020YFA0406300.

**Data Availability Statement:** Not applicable.

**Conflicts of Interest:** The authors declare no conflict of interest.

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
