# Peer review of "Performance Study of a New Cluster Splitting Algorithm for the Reconstruction of PANDA EMC Data"

_instruments, doi:10.3390/instruments6030034_

Round 1

Reviewer 1 Report

General content:

The MC sample used to model the lateral shower development is not sufficiently described. Did the authors use samples at different angles to check that the model is independent of it ? For example, since crystals have different cross-angle in the longitudinal direction, the model function could depend on the angle. Did the authors check that the model does not depend on the pion momentum?

Similarly, the conclusions about the performance of the cluster-splitting algorithm are drawn from a MC sample. Is this the very same sample used to model ? Does the performance depend on the angle ? What is the effect of the new splitting procedure on the shower position resolution ?

Minor English comments:

ll 14-15: the sentence is suspended. Remove "which"

ll 22-23: replace the full stop after "plane" with a comma and continue the sentence with "while"

l 23: contains -> contain

l 25: start sentence with "The"

l 26: the EMS is meant to measure the energy of the photons and electrons, not their energy "deposition in material". Replace words in quotes with "of"

ll 36-39: energy and momentum are used interchangeably. In this sentences what matters is the momentum (doesn't matter if they are "almost" the same)

l 40: remove "by" and add a comma

l 41: of lateral -> of the lateral; considering -> taking into account

l 45: forming -> formed

l 49: to do the separation -> intended to separate

Section 3: line numbers not assigned

second line: remove "as" before colon

in the parenthesis, make it clear what is shown by the dashed line is the function - from the text is sounds like the constant...

seventh line: "with MC simulated events is performed" -> is performed using MC simulated events.  End with a full stop then start a new sentence "The results of the simulation are shown BY the scattered...etc.

Tenth line (beginning of second paragraph): measurement -> model

Following line: "center is not coincide" -> center DOES not coincide

Figure 5 caption: the case that -> a case where - Then the sentence needs LOTS of articles everywhere: THE center of THE seed crystal, THE target crystal, THE seed crystal, THE center of THE shower

L. 65: "small-cross-angle" which angle is meant here ? The opening angle of the photon pair of the "theta" angle at which the photon is emitted ?

Author Response

Dear reviewer,

Thank you very much for reviewing the paper. All the comments are considered in the updated version. Please see the attachment.

Best regards,

Guang

Reviewer 2 Report

The paper offers a new description of the lateral electromagnetic shower profile in terms of energy deposited in the EMCal crystals. This is used to improve the performance of the cluster splitting algorithm to define the energy and position of the overlapping showers in the EMCal. The proposed parameterization can be utilized in many other EMCals with (nearly) projective geometry. Therefore I think it would be important to mention explicitly in the text that PANDA EMCals are approximately projective (with crystal axis looking into the collision vertex). Also, it would be useful to mention the concrete values of the parameters p1, p2, p3 and p4, for example for the case shown in Fig.4.

Below are a few suggestions:

Line 4: "the classical cluster splitting algorithm" - I would not say so, people for many decades have been using more realistic description (not a simple exponent). I'd rather call it "the simplest cluster splitting algorithm".

Line 14: remove "which"

Line 26: I'd suggest you replace "energy deposition in material by the photons and electrons" with "energy of the photons and electrons". I think this would sound better and more correct (note "material" may also mean material on the way to the EMCal).

Line 31: Sounds like each slice has a "lateral size of 21.3mm". I guess this size relates to a crystal. So, the sentence needs to be recomposed.

Line 32: "with different cross-angle with respect to the beam axis": I think here is the place to explain why that - I guess it is to get a (approximately) projective geometry for EMCal crystal orientation. 

Line 45: "forming" -> "formed"

Author Response

(The authors gave the same response as above.)
